

# Exploring the behavioral reactions to a mirror in the nocturnal grey mouse lemur: sex differences in avoidance

Pauline B. Zablocki-Thomas[1,2,*], Grégoire Boulinguez-Ambroise[1,*], Camille Pacou[1], Justine Mézier[1], Anthony Herrel[1,3], Fabienne Aujard[1] and Emmanuelle Pouydebat[1]

[1] Département d'Écologie et de Gestion de la Biodiversité, Muséum national d'Histoire Naturelle, Paris, France
[2] California National Primate Research Center, Davis, CA, United States of America
[3] Evolutionary Morphology of Vertebrates, Ghent University, Ghent, Belgium
[*] These authors contributed equally to this work.

## ABSTRACT

Most mirror-image stimulation studies (MIS) have been conducted on social and diurnal animals in order to explore self-recognition, social responses, and personality traits. Small, nocturnal mammals are difficult to study in the wild and are under-represented in experimental behavioral studies. In this pilot study, we explored the behavioral reaction of a small nocturnal solitary forager—the grey mouse lemur (*Microcebus murinus*)—an emergent animal model in captivity. We assessed whether MIS can be used to detect a repeatable behavioral reaction, whether individuals will present a similar reaction toward a conspecific and the mirror, and whether males and females respond similarly. We tested 12 individuals (six males and six females) twice in three different contexts: with a mirror, with a live conspecific, and with a white board as a neutral control. We detected significant repeatability for the activity component of the behavioral reaction. There was a significant effect of the context and the interaction between presentation context and sex for avoidance during the first session for males but not for females. Males avoided the mirror more than they avoided a live conspecific. This pilot study opens a discussion on the behavioral differences between males and females regarding social interactions and reproduction in the nocturnal solitary species, and suggests that males are more sensitive to context of stimulation than females.

# INTRODUCTION

Mirror-image stimulation (MIS) and mirror tests are used to study multiple questions in the study of animal behavior. The most common and well-known use is to determine whether animals are capable of self-recognition (*Gallup, 1968*), i.e., whether they display a self-oriented response to a mark made on a body part that is only visible in a mirror (i.e., mark test). Such self-engaged response has been tested in a large range of species over the past decades, including Asian elephants (*Elephas maximus)*, marmots (*Marmota flaviventris*), chimpanzees (*Pan troglodytes*), corvids (*Pica pica*), dolphins (*Tursiops truncatus*), and ants (*Formicidae*) (*Plotnik, De Waal & Reiss, 2006*; *Svendsen & Armitage, 1973*; *Gallup, 1970*;

Corresponding author
Grégoire Boulinguez-Ambroise,
gregoire.boulinguez-ambroise@cri-paris.org

*Prior, Schwarz & Güntürkun, 2008*; *Reiss & Marino, 2001*; *Cammaerts Tricot & Cammaerts, 2015*). The results brought out that many species are presenting social behaviors as a response to the mirror presentation, including aggressive behaviors (e.g., tooth chatter and lunges in marmots; jaw-clapping and charging in dolphins). Yet, only a handful of species appears capable of self-recognition, specifically apes species (*Gallup Jr, Anderson & Shillito, 2002*), killer (*Orcinus orca*) and false killer (*Pseudorca crassidens*) whales (*Delfour & Marten, 2001*), and Asian elephants (*Plotnik, De Waal & Reiss, 2006*). However, these conclusions remain debated in particular for non-primates (*Brandl, 2018*; *Gallup & Anderson, 2018*). Besides describing self-recognition (*Heyes, 1994*), MIS are also a relevant tool to investigate whether species behave socially toward the image of a "conspecific" (*Becker, Watson & Ward, 1999*).

Individuals can be more or less attracted by the image of a conspecific, which depends on how they look and how they behave (*Armitage, 1986*; *Réale et al., 2007*; *Cattelan et al., 2017*). Using a mirror instead of a conspecific has the advantage of standardizing the reaction of the conspecific image, while removing most acoustic and olfactory cues (*Svendsen & Armitage, 1973*). For instance, in titi monkeys (*Plecturocebus cupreus*), a study showed that using MIS successfully simulated the presence of conspecifics, suggesting that the use of MIS could be also useful in other primate species (*Fisher-Phelps et al., 2016*). Mirror tests are relevant experiments to investigate personality traits as well (*Réale et al., 2007*; *Wilson et al., 2011*). Specifically, mirror tests are used to describe the aggression and sociability axis, and have been extensively used in marmots (*Marmota flaviventer*) and squirrels (*Sciurus carolinensis, Sciurus vulgaris*) (*Svendsen & Armitage, 1973*; *Armitage, 1986*; *Wauters et al., 2019*; *Petelle, Martin & Blumstein, 2019*; *Santicchia et al., 2020*). Research on animal personality aims to understand why some individuals respond consistently in a different manner compared to others in a given context. Personality traits can be quantified for each individual by carrying out specific tests. For example, the largely used open field tests and novel-object tests provide data on exploration behavior, activity or boldness (*Dammhahn, 2012*; *Dammhahn & Almeling, 2012*; *Highcock & Carter, 2014*; *Perals et al., 2017*; *Savidge & Bales, 2020*), but other experiments like handling (*Verdolin & Harper, 2013*) and tonic immobility (*Erhard & Mendl, 1999*) are used to quantify aggressiveness in various taxa. Aggression and sociability, as personality traits, have also been addressed using mirror tests; for example, in yellow-bellied marmots (*Marmota flaviventris*; *Svendsen & Armitage, 1973*; *Armitage, 1986*) or in rainbow kribs (*Pelvicachromis pulcher*; *Scherer, Buck & Schuett, 2016*), making MIS a valuable tool in this area of investigation as well.

Although a wide range of taxa have been tested using MIS (i.e., facing a full-sized mirror in their environment, either in the wild or captivity), predominantly diurnal and social species have been tested to date. These include fish (guppies, *Poecilia reticulata*; *Cattelan et al., 2017*; Siamese fighting fish, *Betta splendens*; *Takeuchi et al., 2010*), molluscs (*Sepia officinalis*; *Palmer, Calvé & Adamo, 2006*), birds (magpies, *Pica pica*; *Prior, Schwarz & Güntürkun, 2008*), insects (ants, *Hymenoptera, Formicidae*; *Cammaerts Tricot & Cammaerts, 2015*), as well as numerous mammals (Asian elephants; *Plotnik, De Waal & Reiss, 2006*; marmots; *Svendsen & Armitage, 1973*; great apes; *Gallup, 1970*; *Westergaard & Hyatt, 1994*; *Hanazuka et al., 2018*; marine mammals, *Tursiops truncatus*; *Reiss & Marino, 2001*; but also in the

solitary giant panda, *Ailuropoda melanoleuca*; *Ma et al., 2015*). The only MIS study to date using strepsirrhines focused on a nocturnal species: the Garnett's greater bush baby (*Otolemur garnettii*; *Becker, Watson & Ward, 1999*). When close to the mirror, and oriented to it, bush babies showed threat gestures and bipedal postures. Moreover, males were found to engage more in scent marking than females in immediate proximity to the mirror (*Becker, Watson & Ward, 1999*). Thus, the authors of the study concluded that, even if nocturnal strepsirrhines rely more on olfactory and auditory stimuli for social communication, they also appear to be sensitive to mirror-image simulations (*Becker, Watson & Ward, 1999*), as compared to diurnal species that rely predominantly on vision for communication (*Doyle & Martin, 1979*; *Siemers et al., 2007*; *DelBarco Trillo & Drea, 2014*; *Drea, 2020*).

Here, we present data on MIS on the nocturnal grey mouse lemur (*Microcebus murinus*). This species lives in dry deciduous forests along the western and southern coasts of Madagascar (*Mittermeier et al., 2010*). Although this small lemuriform (i.e., about 80g) forages alone during the night, it is not solitary and shows a complex and yet poorly understood social structure (*Radespiel, 2000*; *Schwab, 2000*; *Weidt et al., 2004*; *Dammhahn & Kappeler, 2005*). While mouse lemurs rely mainly on auditory and olfactory cues for social communication, their visual system is well developed and used when foraging in the complex, discontinuous 3D-arboreal environment (*Pariente, 1979*; *Schilling, 2000*; *Ho et al., 2020*). Visual cues are also used for social communication (*Petter, Albignac & Rumpler, 1977*; *Schilling, 2000*); for instance: tail-lashing (moving tail from side to side) and mouth opening are used to threaten, folded ears and tail-curling are used to express fear, and genitalia presentation is used in a sexual context (female sexual behavior). During the day, *M. murinus* shows gregariousness in tree holes. Sleeping groups constitute the basal social unit and can contain up to 16 individuals (*Radespiel et al., 2001*; *Weidt et al., 2004*). It has been reported that sleeping groups are stable, as well as home ranges, suggesting stable social relationships and individual recognition (*Radespiel, 2000*; *Weidt et al., 2004*). Female sleeping groups are usually made up of closely related individuals, showing a matrilineal social structure (*Radespiel et al., 2001*). Indeed, sleeping groups seem to be mostly made of individuals of the same sex, but this changes during the mating season (*Martin, 1973*). Mouse lemurs have one breeding season a year and females can have more than one estrus cycle per season (*Blanco, 2008*; *Wrogemann, Radespiel & Zimmermann, 2001*). Like most lemuriforms, females are dominant over males (*Kappeler, 1993*) and present an increased aggressiveness towards the male solicitations before becoming receptive (*Gomez et al., 2012*). Despite these elements, the social life of this nocturnal species remains poorly understood. A better understanding of its social behavior would be of interest and might provide insights into the evolution of sex differences, but also, it may further help improve breeding and maintenance of the species in captivity (*Languille et al., 2012*; *Roberts, 2019*).

In this study, we aimed to compare the response of *M. murinus* to a MIS and to a live conspecific, as well as assessing possible differences between sexes in their social response. We quantified several behaviors related to activity, approach, and vigilance. We tested whether behaviors were consistent across time and contexts within individuals, and tested for possible sex-related differences in behavior. We predicted that (1) this experiment will allow for the detection of repeatable reactions across time and context within individuals,

(2) individuals will present a similar reaction toward a conspecific and the mirror, (3) males and females will differ their responses in the mirror context (Mir). The results of this pilot study will allow us to assess whether MIS is a relevant tool to investigate social behavior in *M. murinus*.

## MATERIALS & METHODS

### Subjects

We studied twelves (six males and six females) captive grey mouse lemurs (*Microcebus murinus*) that were born and raised in the colony of the UMR 7179 (CNRS/MNHN) of the Muséum national d'Histoire Naturelle (Brunoy, France, Agreement F91-114-1). We conducted the study in July and August 2018. Individuals lived in groups of two to seven individuals in large aviaries ($167 \times 60 \times 70$ cm), maintained under artificial light conditions, allowing control over season and the period of day (photoperiod). Besides, the aviaries were enriched with fresh leafy branches and wooden nest boxes, while their temperature was maintained around 25 °C and the humidity around 30%. Food and water were available *ad libitum* (including fresh fruits, a milky mixture, and mealworms). The behavioral tests took place after the reproductive season, so that females were not in estrus. Animals were individually identified via ear tags.

### Behavioral tests

We built two behavioral test boxes made out of Plexiglas ($40 \times 25 \times 25$ cm), each with one lateral panel and a transparent front panel. A mirror ($20 \times 20$ cm) or an opaque board could be placed in front of the transparent panel (Fig. 1). The boxes were divided into three equally sized zones, separated by an imaginary line: Zone 1 with the transparent panel/mirror/opaque board, Zone 2 in the middle, and Zone 3 at the opposite side of the box. The size of the mirror allowed the individual to see its whole body.

After removing the subject from its individual nest box, we opened the test box and placed the subject in Zone 3. The trial started when closing the test box. The behavioral experiment lasted 30 min and consisted of letting the subject move freely in the test box in three contexts: (1) with a mirror (Mir); (2) with lateral panel opaque as a neutral control context (OF); and (3) a live control context (T), where another individual of the same sex is visible through the transparent Plexiglas. For the T context, a live conspecific from the same age and sex class was in an identical box adjacent to the subject's box. Individuals that were tested together were not familiar to each other, meaning that they did not live in the same aviary. This context (i.e., conspecific) allowed us to collect data on two animals at the same time to maximize data collection. We selected a sample including pairs of individuals of same sex, age and corpulence for this purpose. We recorded behavioral tests using a digital camera (SONY Handycam DCR-SR75) with the infrared (IR) mode activated, as we filmed in the dark. We added an IR spotlight to gain visibility. We proceeded to a blind analysis (meaning that the scorer did not know the identity or sex of the individual) to quantify behaviors using the software VLC media player.

During the three contexts, we recorded eight behavioral variables related to activity, approach, and vigilance (Table 1). Individuals were tested twice in each context for a total

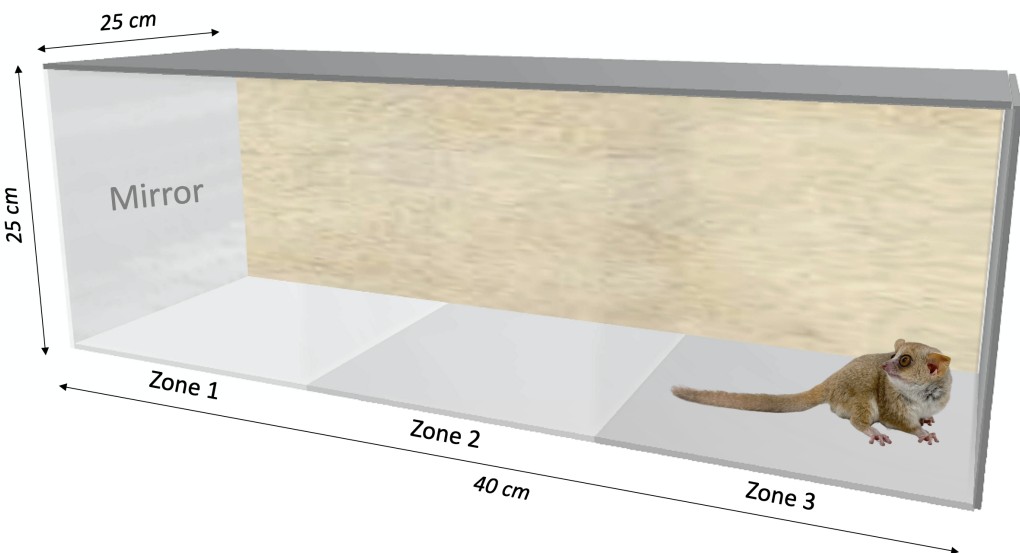

**Figure 1 Behavioral testing box (40 × 25 × 25 cm) used for the mirror-image stimulation study in *Microcebus murinus*.** Three zones were delimited to quantify the behavioral reaction. The sidewall of zone 1 is either transparent facing the other box, a mirror (20 × 20 cm), or an opaque board, depending on the test context. The animal enters the test area in zone 3.

**Table 1 Description of the quantified behavioral variables.**

| Behavioral variable | Description |
| --- | --- |
| Latency to first movement | Time elapsed before crossing from zone 3 to zone 2. |
| Number of changes of zones | The number of times moving from one zone to another. |
| Time spent rearing | Time during which both hands are off the floor. |
| Latency to approach | Time spent before entering zone 1. |
| Number of approaches | The number of times entered zone 1. |
| Number of mirror contact | The number of touches to the mirror with any body part. |
| Time spent in the mirror cell | Time spent in zone 1. |
| Number of rearings | The number of times standing with both hands off the floor. |

of six tests per individual, in random order (the three contexts for the first session, and after that the three contexts for the second session were randomized each time). We conducted repetitions within an interval of 10 to 22 days. However, two females did not participate in all six tests because of health issues: one female experienced Mir and T only once, and another female was run in T only once. We kept these records as NA for the analysis.

## Statistical analysis

We first conducted a principal component analysis (PCA) on the eight variables and subsequently conducted a VARIMAX rotation on the recorded behaviors to reduce the number of behavioral variables to a smaller number of independent variables (*Budaev, 2010*). The first three principal components –named PC1, PC2 and PC3- accounted for
47.3%, 13.4% and 27.0% of the behavioral variance, respectively. We kept only three components as jointly they explained more than 80% of the total variability. We used the acp.varimax() function provided by "psych" package (*Revelle, 2019*) with R software version 3.6.0 (Rstudio Version 1.1.456).

Subsequently, we estimated principal component repeatability as an intraclass correlation coefficient, which is the ratio of the interindividual variance to the sum of inter- and intraindividual variances (*Nakagawa & Schielzeth, 2010*). For that purpose, we ran a linear mixed effects model, with only individual as random effect and no fixed effect. We used the rpt() function in the "rptR package" (*Nakagawa & Schielzeth, 2010*).

We used a non-parametric two-way ANOVA with context as a repeated factor and sex as an independent factor. We used the package nparLD and the f1.ld.f1 function (*Noguchi et al., 2012*). When the ANOVA was significant, we conducted a Friedman test to detect between which test contexts a difference was observed (for both sexes, and for females and males separately) followed by a post-hoc correction with the pgirmess package (*Giraudoux et al., 2018*) and the friedmanmc function (proceeding to pairwise comparison followed by the observed difference between the rank sums (*Sidney, 1957*)). We used $\alpha = 0.05$ as the threshold for statistical significance. Boxplots were drawn in R using the geom_boxplot() function provided by ggplot2 package (*Wickham, 2016*).

### Ethical note

All subjects included in the study were born and reared in captivity in the colony of the UMR 7179 (CNRS/MNHN) of the Muséum National d'Histoire Naturelle (MNHN, Brunoy, France, Agreement F91-114-1). Observations and animal handling were performed in accordance with the relevant MNHN guidelines and the European Union regulations (Directive 2010/63/EU).

## RESULTS

The first principal component (PC1) loaded positively with activity-related behaviors: high values of PC1 correspond to a higher number of moves, a greater occurrence and duration of rearing, a higher occurrence of approach and mirror contacts. High values of PC3 correspond to a longer latency to start exploration and to approach the mirror. By contrast, the second component (PC2) was highly positively determined by time spent in the mirror zone (Zone 1), meaning that individuals with a low PC2 value avoided Zone 1 more than individuals with high PC2 scores (Table 2). PC1 can be interpreted as an "activity" score, PC2 as an "approach" score, and PC3 as an "anxiety" score. Next, we tested for repeatability across the three contexts and sessions (two sessions for three tests contexts) using the principal components scores for each individual. Only PC1 was repeatable (Table 3).

As PC1 was repeatable, we averaged it over the two sessions by individual to have a more representative and unique estimator of activity. For PC2 and PC3, we kept only data from each subject's first session. The repeated measures ANOVA on the first principal component score for each individual (PC1) revealed no effect of sex ($P > 0.40$, $df = 1$, $F = 0.69$) or context ($P > 0.92$, $df = 1.9$, $F = 0.07$), and no interaction between the two

**Table 2  PCA loadings of behaviors.**

|  | PC1 | PC2 | PC3 |
|---|---|---|---|
| Latency to First movement | −0.38 |  | **0.90** |
| Number of changes of zone | **0.87** |  | −0.37 |
| Time spent rearing | **0.78** |  | −0.33 |
| Latency to approach | −0.43 |  | **0.88** |
| Number of approaches | **0.87** |  | −0.38 |
| Number of mirror contact | **0.78** | 0.29 | −0.19 |
| Time spent in the mirror cell |  | **0.97** |  |
| Number of rearing | **0.85** |  | −0.375 |

Notes.
   Variables loading strong on each axis are indicated in bold.

**Table 3  Summary of the repeatability tests on the three principal components extracted from behavioral variables.**

|  | R | SE | CI | P |
|---|---|---|---|---|
| PC1 | 0.288 | 0.128 | [0.023, 0.523] | 0.003 |
| PC2 | 0.12 | 0.102 | [0, 0.361] | 0.085 |
| PC3 | 0.034 | 0.068 | [0, 0.216] | 0.296 |

factors ($P > 0.30$, $df = 1.9$, $F = 1.21$). For PC3 (Session 1), there was also no effect of sex ($P > 0.98$, $df = 1$, $F = 0.0008$) or context ($P > 0.96$, $df = 1.9$, $F = 0.038$) and no interaction between the two factors ($P > 0.82$, $df = 1.9$, $F = 0.18$). For PC2 (Session 1), although we found no effect of sex ($P > 0.49$, $df = 1$, $F = 0.49$), we found a significant effect of context ($P = 0.009$, $df = 1.3$, $F = 5.79$), with a higher approach score in the T context (with a conspecific) and a lower approach score in the Mir (mirror) context (Fig. 2). In addition, an interaction between the two factors ($P = 0.03$, $df = 1.3$, $F = 4.01$) was detected, with males having a higher approach score (higher PC2) in the T context compared to the Mir context, whereas females spent less time in Zone 1 in Mir and T contexts compared to the OF context (Fig. 2).

   Finally, using a Friedman test followed by a post-hoc correction on PC2 for males and females together, and for males only, we found a significant effect of the test context, with individuals spending more time in Zone 1 during the context with the conspecific than with the mirror (all individuals: Friedman chi-squared = 7.09, $df = 2$, $P < 0.029$; males: Friedman chi-squared = 10, $df = 2$, $P < 0.0067$). There was no difference between the three contexts for females only (Friedman chi-squared = 1.33, $df = 2$, $P = 0.51$).

## DISCUSSION

Our results showed a significant repeatability for "activity-related behaviors". Indeed PC1, that mainly showed strong loadings for body movements and displacements, was repeatable across time and contexts, and could thus be considered as related to overall "activity". PC2 was mostly determined by the time spent in Zone 1 (i.e., close to the mirror/conspecific) and was interpreted as an "approach score", while PC3 was mostly determined by the

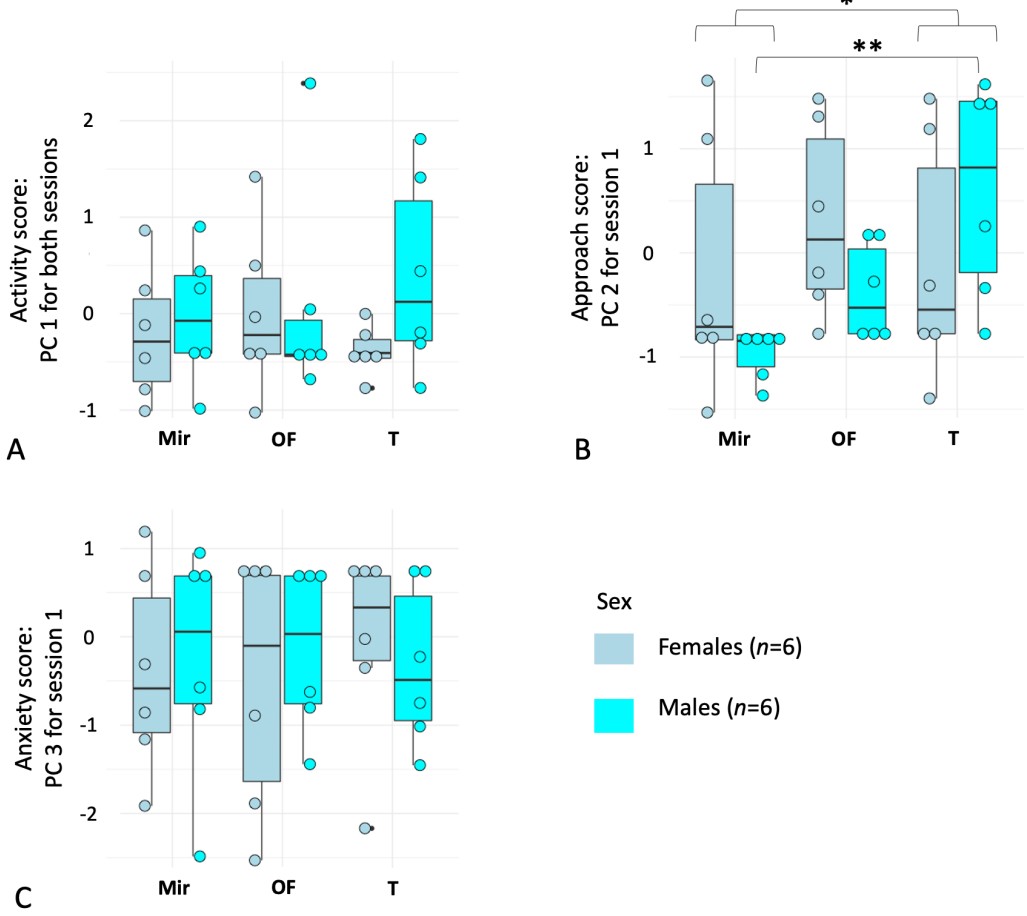

**Figure 2** **Boxplots of principal component analyses run on behavioral variables quantifying reaction to mirror-image stimulations in female ($n = 6$) and male ($n = 6$) mouse lemurs (*Microcebus murinus*).** The figure shows (A) Principal Component (PC) 1 related to "activity", (B) PC2 which can be described as an "approach score", and (C) PC3 which can be described as an "anxiety score", separated by sex and test context. The contexts were: (Mir) mirror-image stimulation, the mirror visible; (OF) a neutral control with an opaque and lateral panel; (T) a positive control context (i.e., facing a conspecific). Each point represents an individual. The middle line of the boxplot represents the median. The lower and upper hinges correspond to the first and third quartiles (the 25th and 75th percentiles). The upper and lower whisker extends from the hinge to the largest and smallest value no further than 1.5 * IQR from the hinge (where IQR is the inter-quartile range, or distance between the first and third quartiles). *P*-value * < 0.05; ** < 0.01.

latency to start moving and can be described as an "anxiety score". Both PC2 and PC3 were not significantly repeatable. In males, but not in females, we detected a differential approach score between the mirror and the live conspecific. Only one case of an attack toward a mirror was reported in a female during a preliminary test (personal observation).

We predicted repeatable reactions across time and contexts within individuals. In fact, we found the first component (i.e., PC1, "activity score") to be repeatable, being in line with previous studies on captive and wild *M. murinus* that have reported significant but moderate repeatability for behavioral traits: 0.33 for emergence latency, 0.28 for

agitation, 0.25 for latency to start exploration in an open field test (*Thomas et al., 2016*; *Zablocki-Thomas et al., 2018*). Consequently, we suggest that MIS could be a useful tool in understanding personality in this species. Conducting other behavioral tests would be necessary to be able to integrate our "activity score" to the animal personality framework. However, this personality axis related to activity appears to be relatively easily detectable in *M. murinus* (*Dammhahn, 2012*). Sociability is less easy to quantify in this solitary forager since it is active during the night and thus more difficult to observe than diurnal species. We used MIS to attempt to detect within-individual repeatable behaviors linked with sociability across time in this species as it has previously been done in other mammal species (*Svendsen & Armitage, 1973*; *Armitage, 1986*; *Wauters et al., 2019*; *Petelle, Martin & Blumstein, 2019*; *Santicchia et al., 2020*). However, we did not detect repeatable social behaviors such as approaching the mirror/conspecific or scent marking as documented for bush babies (*Becker, Watson & Ward, 1999*).

Our second prediction was that the individuals would present a similar reaction to the mirror and to the conspecific. We compared reactions to a mirror with those to a live conspecific, which could either elicit a similar reaction because of the visual similarity with the image, or a different reaction because of the lack of auditory or olfactory cues (*Svendsen & Armitage, 1973*). In our study, we found no significant difference in the composite behaviors PC1 (i.e., activity) and PC3 (i.e., anxiety) of *M. murinus* when presenting either a mirror, a conspecific, or an opaque wall. However, there was a significant effect of context and a significant interaction between sex and context for PC2 (i.e., the approach score): (1) overall, compared to the opaque wall, individuals approached the conspecific and avoided the mirror, and (2) males responded differently to the conspecific compared to the mirror, showing more approaches to the former. We did not expect to find such differences between the mirror context and the positive control context (i.e., with conspecific) in males, since mirror tests are supposed to simulate the presence of a conspecific (*Svendsen & Armitage, 1973*). Our finding is not consistent with a study on West African cichlids, which posits that mirror and conspecific presentation should elicit comparable responses (*Scherer, Buck & Schuett, 2016*). However, *Cattelan et al. (2017)* tested the accuracy of the comparison of mirror tests to tests with living conspecifics in guppies (*Poecilia reticulata*). They measured sociability in both contexts, and showed that the MIS may be less reliable than tests with conspecifics if some improvements were not made to simulate more naturalistic situations (*Cattelan et al., 2017*). Other studies have correlated the aggressive response to a mirror with the aggressive response in tests with a conspecific (e.g., *Scherer, Buck & Schuett, 2016*). However, animals tended to be more aggressive toward the conspecific than to mirror images. Altogether, these studies seem to show that the behavioral response toward a mirror compared to other social/control stimulation is species-specific and presumably dependent on the ecology of the animal (e.g., social system, territoriality, communication, habitat use). The fact that nocturnal mammals rely mostly on olfactory and auditory cues for social communication does not appear to explain the difference in the reaction across contexts. Other nocturnal animals tended to show differences to social video stimulations versus non-social stimulations. For example, rats spent more time in the presence of a mirror or a video image than in an empty chamber without image stimulation, showing that

these animals can visually discriminate between the absence and the simulated presence of a conspecific, and respond as if they were in presence of a conspecific (*Yakura et al., 2018*).

Our unexpected results on the reaction toward a conspecific and a mirror in *M. murinus* suggest that the MIS should be validated and studied extensively for each species, as was previously suggested for fish (*Balzarini et al., 2014*). Other studies using MIS conducted with non-primate mammals (marmots (*Marmota flaviventer*) and squirrels (*Sciurus carolinensis, Sciurus vulgaris*) (*Svendsen & Armitage, 1973*; *Armitage, 1986*; *Wauters et al., 2019*; *Petelle, Martin & Blumstein, 2019*; *Santicchia et al., 2020*)) use this approach as a routine tool to assess personality. For example, in *Wauters et al. (2019)*, the authors use MIS to show that personality in squirrels is impacted by ecological competition between species. In primates, the majority of the studies focus on the theory of mind (*De Veer & Van den Bos, 1999*) and do not attempt to demonstrate that MIS can be used to simulate the presence of a conspecific. In captive titi monkeys (*Plecturocebus cupreus*), where the absence of self-recognition has been acknowledged (*Fisher-Phelps et al., 2016*), MIS represents a valuable tool too to study social relationships. In this species (*Mercier, Witczak & Bales, 2020*), mirror presentation elicits reactions similar to the presentation of a couple of conspecific strangers (e.g., coordinated tail-lashing within pair-mates).

Interestingly, we found that males responded differently to the conspecific compared to the mirror, while female behaviors did not change significantly across contexts, confirming our third prediction. Our results are not fully consistent with a report on male bush babies (*Becker, Watson & Ward, 1999*) approaching the mirror more than females. While we also found a difference between sexes, our results suggest that males distinguished between the mirror and the live conspecific, whereas females did not, with males avoiding the mirror. We cannot fully compare our study with the study on bush babies since they did not use a context with a living conspecific. Here, we can only extrapolate our result on captive to wild *M. murinus*. From an ecological perspective, a difference between male and female *M. murinus* in the reaction to the mirror and to a conspecific could be interpreted as a difference in behavioral strategy between the sexes (*Eberle & Kappeler, 2004a*; *Eberle & Kappeler, 2004b*). Females, for example, sleep in larger groups with related individuals whereas males are more solitary (*Radespiel et al., 2001*). In this species, social interactions between sexes during the active phase are rare and typically occur only once a year during the reproductive season (*Perret, 1997*; *Andrès, Solignac & Perret, 2003*), when the female is sexually receptive for approximately 24 h. Adult male *M. murinus* (over 3 years of age) are bolder than females in the wild, which could be explained by a higher propensity to take risks when looking for a mate (*Dammhahn, 2012*). Besides, avoidance is a submissive behavior and female are dominant over males (*Microcebus sp.*), so that spatial conflicts are most of the time won by females (*Radespiel & Zimmermann, 2001*). This may explain why females present less difference between the three contexts in our experiment. Further investigations during the reproductive season would be relevant, as during this period, individuals are actively looking for mates and are more sensitive to social stimulation. Alternatively, male *M. murinus* avoiding the zone close to the mirror could suggest their incapacity to interpret the simulated presence of a conspecific in a mirror.

We know of no other published MIS experiments to study behavioral reaction differences between the sexes, but several studies involving open field tests, or novel object tests have been conducted and linked to variation in age, sex, and morphology in wild and captive *M. murinus* (*Dammhahn, 2012*; *Dammhahn & Almeling, 2012*; *Thomas et al., 2016*). In the wild (*Dammhahn, 2012*) young (<3 years) male *M. murinus* are bolder than older males and females since they approach novel objects faster and for longer, showing an effect of age and sex on *M. murinus* reactions. In captivity, lighter and smaller individuals were the ones that started exploring new environments showing an effect of morphology on exploration behavior (*Thomas et al., 2016*). In the present study, we could not investigate the effect of age considering the limited sample size of our study. The large variation of behaviors found in our data could also be explained by this limited sample size.

Besides, we did not conduct other biological measures than behavioral measurement. For future research, physiological measurements such as stress-levels (e.g., cortisol levels) may be relevant to better describe the sexual dimorphism in the reaction toward the mirror. In addition, as brain imaging studies are now possible and more easily accessible with animals (*Ferris, 2014*; *Maninger et al., 2017*; *Cook et al., 2018*), we may consider using PET scans to further assess whether males and females are interpreting the situation differently (i.e., determining brain areas involved in the different responses).

## CONCLUSIONS

This pilot study provides first data on the relevance of using MIS to study social behaviors in the nocturnal grey mouse lemur (*Microcebus murinus*), for which data on sociality are particularly lacking. Here, we explored whether *M. murinus* responded with the same reaction toward a mirror presentation and in the presence of a conspecific, and assessed potential sex differences. Whereas females responded in the same manner in all contexts, males did not, avoiding the mirror more than a live conspecific, suggesting a contrast in their social interactions. Such findings offer an insight on sex differences in social behaviors in this emergent model, that may help for both designing future research protocols and managing colonies in captivity.

## ACKNOWLEDGEMENTS

We would like to thank Sandrine Gondor-Bazin, Eric Guéton, and Lauriane Dezaire for helping with the daily care of the animals and Isabelle Hardy for the management of the colony database. We warmly thank Hughes Clamouze for helping us build the experimental device.

### Funding

This work was supported by the ENS of Lyon, Karen Bales' lab (UC Davis, CNPRC) and the Fyssen Foundation for their financial support to Pauline B. Zablocki-Thomas. The funders had no role in study design, data collection and analysis, decision to publish, or preparation of the manuscript.

## Grant Disclosures

The following grant information was disclosed by the authors:
ENS of Lyon, Karen Bales' lab (UC Davis, CNPRC).
Fyssen Foundation.

## Competing Interests

The authors declare there are no competing interests.

## Author Contributions

- Pauline B. Zablocki-Thomas and Grégoire Boulinguez-Ambroise conceived and designed the experiments, performed the experiments, analyzed the data, prepared figures and/or tables, authored or reviewed drafts of the paper, and approved the final draft.
- Camille Pacou and Justine Mézier performed the experiments, authored or reviewed drafts of the paper, and approved the final draft.
- Anthony Herrel analyzed the data, authored or reviewed drafts of the paper, and approved the final draft.
- Fabienne Aujard and Emmanuelle Pouydebat conceived and designed the experiments, authored or reviewed drafts of the paper, and approved the final draft.

## Animal Ethics

The following information was supplied relating to ethical approvals (i.e., approving body and any reference numbers):

All methods were performed in accordance with the relevant Muséum national d'Histoire Naturelle (Paris) guidelines. We studied captive grey mouse lemurs (Microcebus murinus) that were born and raised in the colony of the UMR 7179 (CNRS/MNHN) of the Muséum national d'Histoire Naturelle (Brunoy, France, Agreement F91-114-1). The research adhered to the legal requirements of the European Union: Directive 2010/63/EU.

## Data Availability

The datasets supporting this article are available in the Supplemental Files.

## Supplemental Information

Supplemental information for this article can be found online at http://dx.doi.org/10.7717/peerj.11393#supplemental-information.

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
