# Peer review of "Exploring the behavioral reactions to a mirror in the nocturnal grey mouse lemur: sex differences in avoidance"

_PeerJ, doi:10.7717/peerj.11393_

## Round 0.1 · original submission · Major Revisions

Thank you very much for your submission to PeerJ. I have been fortunate to receive three very thoughtful and helpful reviews from experts in the field. All three reviewers have provided detailed feedback on your manuscript that I believe will help you greatly enhance the presentation of your research.

While the reviewers praised your experimental design and the fact that you tested an under-studied species, all three had concerns with your article as currently presented. A common request among all three reviewers is that you should state your study aims more clearly within your article and that you more clearly relate your findings back to a broader theoretical perspective. I agree that both these will help. Additionally, both reviewers 1 and 2 recommend that you either analyze your data in a way that is more appropriate to your data set or provide stronger justification for the approach you have taken. Of course, and as the reviewers note, reanalyzing your data may well change the results you report and the conclusions you draw.

In addition to these core and common concerns the reviewers all provided a number of detailed comments. I request that you respond to each one as you prepare your article for resubmission. Additionally, where reviewers have requested additional information or detail, please provided that in the article itself, as well as in your response to reviewers, as other readers will likely also want to know that information.

I look forward to receiving your revised article.

Reviewer 1 ·

Basic reporting

In general, the manuscript is well written but I think that the first part of the introduction, as well as the description of the aims, could be improved (see specific comment below).

Experimental design

The research questions and the aims should be better addressed. The experimental design does not show serious weakness. However, some extra details in the methods section should be provided (see specific comments below).

Validity of the findings

The statistical analyses should be redone completely (see below). Thus in this state, the validity of the results cannot be evaluated. However, I appreciate that the authors recognize their study as a pilot, and they did not overstate their findings.

Additional comments

In this paper, authors tested the behaviour of the grey mouse lemur in response to a mirror image, a conspecific and a white opaque board (as control). This species is solitary and nocturnal, and the behaviour has been rarely investigated. The use of the mirror to understand the behaviour of this species can be useful in the future. The topic is interesting and the study is potentially beneficial for future researches on these (and similar) animals. However, I have a major concern and a few relatively minor comments/suggestions.
My main concern is about statistical analysis. The guidelines for analysing behavioural data strongly discourage to perform PCA. Firstly PCA should be performed on large dataset. A very general rule is to having sample size at very least three times the number of variables (see Budaev 2010, Ethology). Secondly, variance explained by two of the three PCAs are very too low. PCA analysis tend to inflate factor loadings as low and non-significant correlations can produce high PCA loadings. As expected from the variance explained and factor loadings, the PCAs with the lowest variance explained failed be repeatable. This strongly suggest that these PCAs should not be used. Instead authors choose to perform analysis by using only the data obtained from the first trial, potentially leading to data that is meaningfully interpreted. The authors can read the paper by Budaev (2020, Ethology) for an extensive review on the guidelines for using PCA. Moreover, a Dingemanse and Wright (2020, Ethology) suggest a more appropriate approach to treat variables recorded within the same trial: “A more appropriate approach would arguably be to report the correlation matrix between the different behavioural measures recorded within the same assay (within and among individuals) and then pragmatically take forward one representative behavioural measure that is normally distributed and covaries (i.e., loads) most strongly with the overall PCA component or latent variable estimated (Araya-Ajoy & Dingemanse, 2014; for other alter- native approaches, see Dochtermann & Nelson, 2014)”. Moreover, from the same paper, authors recommend to avoid using variables from the same behavioural test in PCA to infer behavioural syndrome and personality: “[..] a PCA cannot be used to imply that there is a “behavioural syndrome” or “animal personality structure” if the study does not feature repeated measures and separate behavioural assays.”. In contrast, in this study authors states that they used MIS to detect “[…] significant repeatability for the activity axis of personality” (L 51). Overall, for the above reasons, all the analyses should be redone accordingly.
Since the results could profoundly change after revising statistical analyses, my subsequent comments and suggestions will regard mainly introduction and methods.
LL 60- I think the introduction should be reorganized by providing more details on the background. Specifically, the first part of the introduction should firstly focus on the 2 general aims of using the mirror: 1) to verify whether the individuals of a certain species recognize themselves in the mirror and if not 2) to study the behavioural responses to conspecifics by using the mirror to simulate the presence of live conspecifics. The introduction would strongly benefit from clearly stating these 2 different aims.
L62 question marks is sometimes after and sometimes before brackets.
LL 71-72 please notice that some studies were also performed on highly territorial, aggressive species (e.g. Betta splendens).
LL 78-80 I feel this is a crucial point for justifying the entire study and thus this part should be extended. For instance, how does studying the responses to a mirror image could improve our understanding of these animals? Which types of studies could be beneficiated from the use of this methods? Which information can be obtained? I agree with authors that this methodology can be useful with solitary, nocturnal species but I think this point should be addressed more extensively.
LL 97-98 the study by Cattelan and colleagues was not focused on aggressive interactions, but instead, on social interactions and sociability. There are many other studies performed in fish in which the mirror was used to study aggression: Scherer et al 2016; Arnott et al 2016; Elwood et al 2014; Balzarini et al 2014 and others.
LL 99- see my comment above on the first part of the introduction.
LL 134-135 the aim should be better addressed. I think the most important aims here are 1) to understand whether the individuals of this species recognize themselves in the mirror and if not 2) to investigate whether the mirror can be used instead of a live conspecific to measure behaviour in a more standardized and easier way. Moreover, as I have previously stated, authors cannot infer personality traits/axes by performing a single behavioural assay. Accordingly, authors should rephrase here and throughout the manuscript using “behaviour” and/or “behavioural trait” instead of “personality”/ “personality trait/axis”.
L 136 here authors should explain the term “sociability” more in depth.
LL 137-140 all these behaviours should be detailly explained. For instance, which is the biological meaning for “standing bipedally”? Aggressive posture? Social posture? For a reader not familiar with primates or mammals can be difficult why you are measuring a certain behaviour.
LL 143-144 I do not understand why you should expect that males and females behave differently. Please clearly state your hypothesis.
LL 172-173 did the two individuals were familiar or unfamiliar before the test? This is an important point as familiar individuals can behave differently from unfamiliars.
LL 174-175 please be aware that considering this test as two different tests you are violating the assumption of independence. The behaviour of an individual can affect the behaviour of the other and vice versa.
L 178 “blind” in respect to what?
LL 180-181 please state at which distance of time the two tests were performed.
LL 285 I think that the response strongly depends also on how the mirror test is performed (e.g. the presence or not of improvements).
LL 302-303 totally agree with this. Unfortunately, with this sample size it is difficult to drive any solid conclusion. And indeed I appreciate that authors did not overstate their results, but instead they recognize the limitations of the study (e.g. LL 328-330).
Figure 1: the figure would greatly benefit from showing the sizes of the apparatus.

Reviewer 2 ·

Basic reporting

In this study a MIS-test was used to study personality in grey mouse lemurs. Captive mouse lemurs (N=12) were confronted either with a mirror or same sex conspecific and an opaque board as a control. Responses were summarized in three principle components representing measures of “activity”, “approach” or “anxiety”. Only the PCA representing “activity” measures was repeatable. Overall there was no sex difference in behavior across experimental conditions for PCA1 and PCA3. However, in one session males avoided the mirror and approached conspecifics more often. Hence, the authors conclude that males are more sensitive than females to the different contexts.

The Introduction covers the most important aspects of MIS-test but needs some re-structuring (see detailed comments below). Also, the authors should elaborate a bit more why they used a MIS-test to examine personality in mouse lemurs and adapt their predictions accordingly.
L 54 Why ecological differences between sexes? They usually occur in the same environment.
L 60ff: MIS studies have originally been conducted to examine self-recognition in the context of self-awareness, e.g. Theory of mind. Therefore, it would be good to address this already in the beginning of the Introduction.
L75ff: It’s correct that nocturnal species are considered to rely more on olfactory or auditory stimuli but the MIS test is not applied to study the relative importance on the usage of visual vs olfactory or acoustic stimuli, therefore it would be good to elaborate why it would be interesting to use a MIS-test to study personality in nocturnal primates. Is the relative importance in the use of sensory modalities important to study personality?
L 78: Most MIS-tests have actually been done in captivity, hence, stating that it is difficult to study nocturnal primates in the wild is not a convincing argument.
L112-117: This has been stated already above, even with the same wording.
L134: here you should elaborate why a MIS-test is particularly suited to study personality in mouse lemurs. Why a MIS-test and not an open-field or novel object test? Explain the advantages to use a MIS-test to study personality in mouse lemurs.
L137-140: this is actually part of the Methods
L 142: explain how you measure exploration in a MIS-test.
L144: should be “… more often than females-…”
L144ff: If the intention was to study sociability with the MIS-test, you should also present your predictions accordingly. Do you assume that males are more sociable than females? If so, why? Also consider differences in social behavior between the sexes, i.e., female sleep in groups whereas males usually sleep alone.

Experimental design

The experimental design is in principle flawless. For the statistical analysis the authors should include test session as additional factor. For the PCA the authors should provide more information on the validity of the PCA (see detailed comments below).
L154: insert in groups of two to seven individuals
L 162ff: Did you conduct the experiments in the same or randomized order? If you run them in the same order include order as an additional factor to control for an order effect.
L172: Was it a familiar or unfamiliar individual?
L 190: Which criteria did you use to extract the factors of the PCA? Did you choose the PCA’s according to their Eigenvalue? If so, see Dingemannse & Wright 2020 and Morton & Altschul 2019 for some critical comments. Please also provide information on the Kaiser-Meyer-Olkin value to understand how well the three components explain the given variation.
Dingemanse, Niels J., and Jonathan Wright. "Criteria for acceptable studies of animal personality and behavioural syndromes." (2020): 865-869.
Morton, F. Blake, and Drew Altschul. "Data reduction analyses of animal behaviour: avoiding Kaiser's criterion and adopting more robust automated methods." Animal Behaviour 149 (2019): 89-95.
L184: Video-Analysis: do you have any measures for inter-observer reliability for the video analyses? Please add them.
L196: You also need to include test session as an independent factor to estimate whether order (first or second) has an influence of the response.
L197: add reference for the package nparLD
L226: Did you also check whether you find the same results when you use only the first values of PCA1 instead of the averaged values? Since you did not average the values of PCA2 and 3, it would be better to treat all values consistently throughout the analyses.
L 233: Here you present only the results of the first session but not for the second. Also, it is not clear whether the analyses for PCA1 and 3 are done only for the first or second or both sessions. As mentioned above it would be important to include session as an independent factor in your analyses.

Validity of the findings

In principle, there were no differences in behavior across the three experimental conditions, especially not between the control and the mirror and conspecific presentation, questioning the validity of the results. Especially, in light of the findings that males appear to avoid the mirror and approach conspecifics only in session one, but results of session two were not provided. So, it might be that males responded basically more strongly to novel stimuli (mirror, conspecific) than females. The results should, hence, be elaborated by adding more comprehensive statistical analyses and discussed more critically.
L252: insert “by” after determined
L 257-260: In the beginning of the sentence you state that activity as a personality trait have been found in wild and captive mouse lemurs but then you refer only to studies in captivity.
L 261: Explain what you mean that sociability is not so “easy” to quantify in a solitary forager.
L272: Here you have to explain why you did not expect to find this difference and discuss this finding regarding the social behavior of the species.
L285: Explain more specifically why it should depend on their ecology, which aspects of tehir ecology?
L287: I don’t think that you can state that nocturnal species are “non-visual”; they have a good vision to explore their environment and may only in comparison to diurnal species rely more on olfactory than visual information.
L 314: Here I would also refer to some studies of wild mouse lemurs, e.g.:
Eberle, M., Kappeler, P.M., 2004. Behav Ecol Sociobiol 57, 77–90.
Eberle, M., Kappeler, P.M., 2004. Behav Ecol Sociobiol 57, 91–100.
L319: Why should female dominance explain that females did not approach the mirror or same sex conspecifics?

Additional comments

Although, MIS-test have not yet been conducted in nocturnal primates and in principle the experimental approach is flawless, the statistical analyses have to be improved and the findings accordingly discussed.

Reviewer 3 ·

Basic reporting

Introduction:
-I commend the authors on including a wide variety of studies (citations) that fall outside of the primate order.
--The aim and relevance/significance of the study is not clearly stated. The manuscript would benefit from the authors clearly and succinctly conveying what the aim(s) of the study are, how their work fills a gap within this field of research, and why it is significant/relevant.
--In regards to the introduction and background, the manuscript would benefit from a more thorough and succinct look at the relevant literature and provide more detailed examples and interpretations, so that their results/conclusions can be better and more clearly linked to former studies in the discussion. It would also be beneficial if the authors discussed some specific primate examples, other than just the one nocturnal species, Garnett’s greater bushbaby
--Line 87-95, I think this information would make more sense at the beginning of the introduction- helping to lay out what this type of research entails, why it's important, etc. Also, important to define terms at the beginning so the reader is aware and able to follow along
--In regards to animal personality, I think it would be important for the authors to include some information on how consistent personality differences can have fitness impacts on an individual
--Becker et al., 1999 is discussed and used in a comparative content in the manuscript, but the results of this work haven't been interpreted, only reported.
--Mirror tests being employed to investigate aggression (as a personality trait) is mentioned in the introduction and the authors cite two studies (yellow-bellied marmots and guppies), but do not discuss any of the findings.
--The authors discuss how grey mouse lemurs are sexually dimorphic (females larger than males) and that females have a greater bite force than males, but do not explain what this may have to do with MIS (potential impacts), or the relevance to this study
--Lines 135-140, seems to be explaining the methods which seems unnecessary in the introduction

Discussion:
-The discussion section does not expand on/provide interpretations of their results in the context/comparison of past studies, what do the results mean (big picture). The couple of studies that are used to compare are fish studies (also need further explanation), it would be beneficial to also discuss their findings in relation to other research on primates.
--Connections to ecology are made in the discussion but it is not clear what this has to do with the results of the study
--The future work that is mentioned, again, it is not clear what the relevance and significance would be

-There are consistent grammatical errors throughout the manuscript. The manuscript would be greatly improved once addressed. Some examples where grammatical errors need to be addressed include lines 55, 81, 83, 95, 99, 101, 106, 111

-Some portions of the text are unclear and ambiguous. Thus, it is not always obvious what the authors are trying to relay to their audience. Some examples include lines 52-54, 54-56, 112 (Are you referring to M. murinus or a different species of mouse lemur? Perhaps just use Microcebus spp., if applicable), 140 (Can the authors clarify what they mean by ‘test’ (e.g., MIS or a statistical analysis), 154, 286 (Did the authors mistakenly put ‘visual cues’ instead of olfactory?), 319-322, 333-336

-There are a few instances of spacing errors in the manuscript. Some examples include lines 62, 127, 202

-There are some inconsistencies with the in-text citations that need to be addressed. Some examples include lines 61-62, 68, 104, 108, 144

-There are a few instances where scientific names need to be italicized. Some examples include lines 369, 421, 430

-Line 75, The authors provide a citation, Becker et al., 1999 in reference to nocturnal strepsirrhines relying predominantly on olfactory and auditory cues , while diurnal species are more reliant on vision. This citation is a quite dated resource. All strepsirrhine primates use olfactory communication (diurnal, cathermeral, and nocturnal), examples from the literature below
--Drea, Christine M. "Socioecological and phylogenetic patterns in the chemical signals of strepsirrhine primates." Animal Behaviour 97 (2014): 249-253.
--Drea, Christine M. "Design, delivery and perception of condition-dependent chemical signals in strepsirrhine primates: implications for human olfactory communication." Philosophical Transactions of the Royal Society B 375.1800 (2020): 20190264.

-Line 66 needs another parentheses after ‘fishes’

-Lines 112-114, This sentence already appears on lines 73 & 74

-Lines 114--117, This sentence already appears on lines 75-77

-Throughout the manuscript the authors use a variation (contexts, situations, conditions, etc.) of words to refer to the three different contexts mouse lemurs are presented with (i.e., mirror, live conspecific, white board). The manuscript would benefit from consistency by using only one word throughout when referring to the three contexts.

-Lines 191-193 contains a quote, the citation does not provide a page number(s)

-Line 217 suggestion to change ‘higher’ to greater

-There are inconsistencies with how the authors refer to principal component analyses (e.g., PC1, (PC) 1). It would be beneficial to employ one way of referring to PCAs.

-There are several inconsistencies with the reference formatting. Some examples include lines 381, 393, 406, 421

-Line 331, can the authors clarify what they mean by “clearly, something different happened in males…”

-Line 131 delete ‘primates’

-The authors report some of their results in the description of Figure 2. It would be beneficial to just include the information a reader would need to interpret the figure.

Experimental design

-The study conducted by the authors fits within the scope of PeerJ

-The authors clearly lay-out their predictions for the study (lines 140-146). However, in regards to the research question, the manuscript would benefit if the authors were to more clearly and succinctly convey what the aim(s) of the study are, how their work fills a gap within this field of research, and why it is significant/relevant.
--For example, the authors state, “This pilot study opens a discussion on ecological differences between males and females in the nocturnal solitary species and suggests males are more sensitive to the context than females” (lines 54-56). It is not totally clear to me what exactly they mean by this. There is no further discussion of what ecological differences they may be referring to.
--The only other mention in the manuscript about ecological differences appears in the discussion (lines 311-313). It is unclear to me what the authors are trying to say (i.e., the connections of ecology, ecological differences between the sexes and this study and its results).

-The authors state that this study is a pilot study (line 54) in the abstract and mention it again at the end of the discussion (line 322). However, this manuscript would benefit if a statement was included at the end of the introduction when the authors lay out what this study will entail.

-While the sample size is small (n=12), the authors recognize this and state this clearly in the manuscript. Additionally, it is not the easiest to find a large colony of captive mouse lemurs in which to perform research!

-The authors' investigation adhered to an ethical standard, which is stated clearly in the manuscript.

-I commend the authors on a detailed and informative methods section, which clearly lays out the steps necessary for future replication.

Validity of the findings

-The authors have provided all of the raw data

-All statistical analyses are robust and statistically sound. The study is scientifically sound and appropriate for PeerJ.

Additional comments

I commend the authors on a well designed study with robust statistical analyses. The figures are of high quality and labeled and described well. I especially love Figure 1! Once the language and grammar issues are addressed, as well as the aforementioned comments on the introduction and discussion, this paper will be ready for publication.

Most important points to be addressed:
-Language and grammar
-Aim and relevance/significance of the study
-How this work fills a gap within this field of research, and why it is significant/relevant.
-Connections to ecology to be clarified
-Expand on/provide interpretations of their results in the context/comparison of past studies, what do the results mean (big picture).

---

## Round 0.2 · Minor Revisions

Two of the three reviewers who reviewed your original submission have now provided feedback on your revised submission. While both thank you for the improvements that you have made to your reporting, both also have further suggestions to help you enhance your article. I encourage you to respond carefully to their suggestions. This includes the potential to run new analyses to consider the effect of different testing intervals.

As noted by one of the reviewers, you report that this is a pilot study only in the final sentence of your Introduction, and then again in your Conclusions section. If you consider this to be a pilot study, which is reasonable, I suggest you present that in a more upfront manner - perhaps even in the title - and that you more explicitly state and discuss the limitations of your study in your Discussion.

Reviewer 1 ·

Basic reporting

The authors improved the introduction and in general the presentation of the study.
Overall I am satisfied with the revisions they mainly did on the introduction and discussion section. The manuscript is much more improved compared to the original version.

Experimental design

After revising the manuscript, the research question and the predictions are well defined.
Moreover, the authors provided more details on the methodologies used, as requested by reviewers. Now the methodologies are fully described.

Validity of the findings

Repeatability was conducted within an interval that ranges between 10 and 22 days. This variation is extremely high and could be affected the results. My suggestion is to formally test the effect of days between the trials.

However, I am not yet fully convinced of using PCA analysis in this study, as I have extensively explained in my previous revision. The problem in using PCA in this study appears evident from the fact that only one out of three PCAs was repeatable.

The results are extensively discussed. All the points raised in the discussion sound reasonable.

Conclusions are well stated and the authors acknowledge the limitations of the study and avoid overstating the results.

Additional comments

L191 add "sexually" before receptive

L 387 remove one "s" in contexts

L 458-459 these scores are in line with those found in other species (in general repeatability for behaviors is quite low).

L 576 correct "sensitive"

Figure 1 I would suggest adding sizes (in for example cm) to the figure.

Figure 2 Adding asterisks to the figure would improve readability.

Reviewer 3 ·

Basic reporting

Abstract:
- There are some grammatical and spacing issues that need to be addressed. For example, Lines: 19, 21, 22, 24
- Lines 23-24, the wording is a bit unclear- ‘…whether mirror presentation elicits a similar reaction than a conspecific…’
- Suggestion to mention that this work is a pilot study earlier in the abstract, perhaps Line 21
- It is still unclear to me how this study/results relates to ecological differences in male and female grey mouse lemurs (Lines 29-31)

Introduction:
- There are some grammatical and spacing issues throughout the introduction that need to be addressed. For example, Lines 34, 37, 42, 46, 48, 76, 79
- Suggestion, Line 35 replace ‘able’ with capable
- Suggestion, delete ‘already’
- Lines 41-44, wording is unclear and confusing, suggestion to delete ‘capable’
- Line 42, can the authors provide a couple examples of ‘social’ and aggressive behaviors’ (e.g., XXXX)
- Line 42, consider adding the species’ names (primates, cetaceans, etc) after ‘only a handful of species…’
- Can the authors clarify what they mean by ‘visual attractiveness’? (Line 48)
- Consider making two sentences instead of one, Lines 48-51
- Suggestion, delete ‘Interestingly’, Line 67
- Lines 77-80, can the authors make a statement of interpretation about the results of the bush baby study. For example, what is or why is it significant that bush babies showed threat and bipedal postures when close to a mirror?
- Lines 80-83, This sentence is unclear and confusing. Why do nocturnal strepsirrhines, which are more reliant on olfactory and auditory cues, ‘appear sensitive to mirror-image simulations’?
- Suggestion, Line 85, delete the first ‘the’ & ‘forest’ and ‘coasts’ should be plural
- Line 93, fix phrasing ‘show threat’
- Line 100, ‘seems’ should be singular
- Line 107-108, can the authors clarify and explain what they mean by ‘…origins of known differences between the sexes’
- Suggestion, Line 109, Change to ‘In this study we aimed to…’
- Suggestion, Line113, Change to ‘We predicted that’, delete ‘that’ after (1)
- Suggestion, Line 113, Change to ‘this experiment will allow for the detection of repeatable reactions across time and space’. Delete period at end of sentence and replace with a comma
- Suggestion, Line 114, delete ‘we also predicted that’, change to ‘individuals will present a similar reaction toward a conspecific and the mirror’. Delete period at the end of the sentence and replace with a comma
- Suggestion, Line 116, delete ‘Finally, we predict’. Change to ‘males and females will differ their responses in the mirror context (Mir).’
- Suggestion, Line 117-118, Delete sentence beginning with ‘Specifically’
- Suggestion, the authors may want to consider mentioning that this study is a pilot study earlier in the introduction, instead of the last sentence

Materials & Methods:
- Suggestion, Line 127, consider changing ‘The room’ to ‘Aviary temperatures were maintained…’ use aviary for consistency instead of room
- Suggestion, Move sentence ‘We tested twelve…’, Lines 129-133 to appear under the Behavioral Tests subheading, perhaps at the beginning of Line 141
- Suggestion, Line 133, consider rephrasing ‘Animals were individually identified via ear tags’.

Behavioral Tests:
-Line 145, add ‘is’, so it reads ‘is visible’
-Line 146, should read ‘age and sex class’
-Line 148, consider changing cage to aviary for consistency

Statistical Analyses:
- Add (PCA) after principal component analysis
- Line 164, consider rephrasing to read ‘The first three…’
- Line 165, delete comma before ‘accounted’
- Line 166, suggestion to add comma after ‘variance’
- Line 169, add comma after ‘Subsequently’
- Line 171, wording is a bit unclear
- Suggestion, delete ‘Then’

Results:
-Line 199, Can the authors clarify what they are referring to when they state, ‘estimator of this behavior’? Are they referring to ‘activity’ as mentioned in Line 194?
-Line 213, add comma after ‘together’
-Line 232-234, The phrasing here is a bit informal. The authors might consider rephrasing this sentence.
-Line 232, Can the authors explain what the possibility of relating their ‘activity score’ to the activity axis of personality would mean or indicate in this context?
-Line 240, delete ‘the’ and ‘of’ and change ‘approach’ to ‘approaching’
- Line 242, delete ‘for example’
- Suggestion, consider citing the paper in-text, Line 256-257
- Line 280, add comma after (2019)
- Line 281, ‘Theory of Mind’ does not need to be capitalized
- Line 285, add comma after ‘In this species’
- Line 285-287, consider rephrasing as this sentence is unclear/confusing
- Line 287, add comma after ‘species’
- Line 288, consider deleting ‘Interestingly’

Discussion:
- Lines 219-227, This paragraph seems like a repeat of the results and not an interpretation of the results reported.
- Line 297, Can the authors clarify what they mean by ‘wild animals’? Are they referring to wild mouse lemurs?
- Line 290-293, The wording here is confusing, consider revising and explain and/or give an example of what is meant by ‘social function’
- Line 300, ‘group’ should be plural
- Line 305-307, This sentence seems out of place
- Line 309-310, consider rephrasing ‘As a follow up to our pilot study…’ and consider revising the wording, as it seems informal. Suggestion to delete ‘interesting to’
- Line 314-316, The wording here is confusing and not clear, consider revising
- Line 317-318, change to ‘between the sexes’
- Line 318, test should be plural

Conclusions:
- Lines 328-340, Language/wording needs rephrasing.
o It is not clear as to what the significance is in relation to the study. The authors state that this pilot study aimed to ‘describe the response of M. murinus toward a mirror presentation and to explore whether they responded as when in the presence of a conspecific”. Why is this important? What is the significance in understanding this? What gap in the field/literature does this fill and/or contribute to?
o How would hormone analyses contribute to the documented reactions of the study subjects? How would they ‘explain sexual dimorphism in the reaction toward the mirror? Same as for PET scans, how does this improve/advance our knowledge?

References:
- Formatting and spacing issues throughout the references. For example, Lines 348, 368, 377, 386, 396

Legends:
- Spacing issues, Line 539
- Line 541, change ‘arena’ to area
- Spacing issue, Line 546
- Line 547, add ‘and’ after ‘panel;’
- Can the authors clarify what they mean by ‘situation’, Line 549. Does this refer to context?

Experimental design

The study fits within the scope of Peer J. The authors’ investigation adhered to an ethical standard, which is stated clearly in the manuscript.

Validity of the findings

The authors have provided all of the raw data and state that their script will be provided freely upon request (see reply to reviewers). All statistical analyses are robust and statistically sound.

Additional comments

I commend the authors on their work addressing reviewer suggestions. However, there are still a number of issues with the current manuscript which need to be addressed.

Most important points to be addressed:
1. Consistent language/wording/phrasing issues throughout the manuscript
2. Grammatical and spacing issues throughout the manuscript
3. Aim/purpose of study is still not clear
4. Discussion- restates the results, there needs to be more interpretation of the results and comparisons of them to other relevant work (to put this work in context)
5. Conclusions- overall conclusions and their significance of the present study is not clearly stated. It is not clear how the ‘future work’ that is suggested would improve/advance knowledge.

---

## Round 0.3 · Minor Revisions

Thank you for responding carefully to the reviewers' last round of comments. I have reviewed your article and responses and do not feel the need to send your article back out for review. However, I did notice some minor errors that needed correcting and ask you to attend to these. When you have addressed these minor comments it will be my pleasure to accept your article for publication in PeerJ.

Line 22 “- the grey mouse lemur (Microcebus murinus) -, an emergent animal” – the additional comma needs deleting

Line 40 the Latin name for chimpanzee is Pan troglodytes, not Pan paniscus (bonobo). Please correct this and check and verify all Latin names throughout.

Line 52 “the image of a conspecific, which depend on” – depend should be depends

Line 56 “a study showed that MIS allows to successfully simulate” suggest rephrasing to “a study showed that using MIS successfully simulated”

Lines 79-82 please check the parentheses used here – I think there are some missing or misplaced around the mammal references that are subset here.

Lines 190 “All methods were performed in accordance with the relevant MNHN guidelines and the European Union regulations (Directive 2010/63/EU).” Please also confirm and state that your study was reviewed by an ethical review body and that approval was granted, providing the approval number for your study if possible.

---

## Round 0.4 · accepted · Accept

Thank you for responding to my final outstanding comments. It is my pleasure to recommend your article for publication.